# Evidence of On-Going Disparate Levels of Care for South Asian Patients with Inflammatory Bowel Disease in the United Kingdom during the Quinquennium 2015–2019

**Affifa Farrukh** * and **John F. Mayberry**

Nuffield Hospital, Leicester LE5 5HY, UK; johnfmayberry@yahoo.co.uk
* Correspondence: Farrukh_affi@yahoo.com

**Abstract:** Over the last decade, there have been a number of studies which have documented disparate levels of care in the management of inflammatory bowel disease amongst various minority communities in the UK. Similar findings had previously been described in the USA, where access to biologics has been an issue. In this study, data on admissions to hospital of South Asian and White British patients with inflammatory bowel disease between 2015 and 2019 were collected from 12 National Health Service (NHS) trusts in England, three Health Boards in Wales and two Scottish health organizations using Freedom of Information requests. The analyses of data were based on the assumption that inflammatory bowel disease (IBD) has the same prevalence in the South Asian community and the White British community in the UK. Comparisons were made between the proportion of hospitalised patients who were South Asian and the proportion who were White British in the local community using a z statistic. In Leicester, Bradford, Croydon and Lothian, the proportion of patients from the South Asian community admitted to hospital was significantly greater than the proportion from the local White British community, which is consistent with the greater frequency and severity of the disease in the South Asian community in the UK. However, in Coventry, Wolverhampton, Walsall, Acute Pennine Trust in the north-west of England, Barking, Havering and Redbridge and Glasgow, South Asian patients were significantly under-represented, indicating significant issues with access to hospital-based healthcare for inflammatory bowel disease. This study provides evidence of on-going evidence of disparate levels of care for patients from a South Asian background, with inflammatory bowel disease being underserved by a number of NHS Trusts, Health Boards and comparable organisations. When there is on-going failure to achieve the objectives of the NHS of achieving equality in the delivery of care, it is critical to introduce effective policies which will alter the in-built inertia to change within such organisations.

**Keywords:** inflammatory bowel disease; South Asian; hospital admissions; disparate care

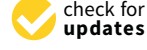



## 1. Introduction

There have been earlier studies that have shown disparate levels of care received by South Asian patients in England [1] and also amongst Eastern European and Afro-Caribbean patients, compared to those who are White British [2]. These studies recognized that the prevalence of inflammatory bowel disease in the South Asian community was comparable to, if not greater than that in the White British population. An underlying problem in remedying such disparate levels of care in the past has been a reluctance amongst managers to take ownership of this issue [3,4]. Indeed, trusts deny the evidence from their own data that such disparate levels of care exist [3]. National organisations, such as the Care Quality Commission and National Health Service (NHS) Improvement, which are tasked with ensuring equitable care, have taken no actions to address issues of disparate levels of care [3]. Indeed, the recent report by the Commission on Race and Ethnic Disparities stated:

"The Commission's view is that individuals and communities of all ethnicities should be encouraged to take control of their own health. This would be both in relation to changing their own behaviours and in taking part in research studies to see what is effective." [5]

Such a view, suggesting that health inequalities are the fault of the communities themselves, reiterates a concept put forward in the 1960s by Patterson, when she wrote no:

"coloured children are deprived of medical care through ignorance, that their basic needs are met by continuous health education and that those who work with coloured immigrants should be trained to understand their problems and to communicate with them." [6]

Or as suggested by John Calmore, they are viewed as being simply natural misfortunes [7].

In this study, Trusts and NHS organisations across England, Scotland and Wales, where there were significant South Asian communities, were selected to investigate the situation more recently between 2015 and 2019. Unfortunately, no relevant data were available from Northern Ireland, with South Asians comprising less than 0.5% of the population. Therefore, the hypothesis under investigation in this study was that admissions to hospital for inflammatory bowel disease should reflect the proportion of South Asians in the local community.

## 2. Method

### 2.1. Participants

In June 2021, a Freedom of Information request was sent to 12 NHS trusts in England, three Health Boards in Wales and two Scottish health organizations (See Table 1). These organisations were selected because they served areas with significant South Asian populations. Included within the group were some that had taken part in earlier studies on the management of inflammatory bowel disease in that community.

**Table 1.** Freedom of Information Requests.

| | | |
|---|---|---|
| University Hospitals of Coventry and Warwickshire NHS Trust | FOI/1096 | Date: 9 November 2021 |
| Barking, Havering & Redbridge University Hospitals NHS Trust | FOI/7457 | Date: 23 September 2021 |
| Pennine Acute Hospitals NHS Trust | FOI/1124 | Date: 22 September 2021 |
| University Hospitals of Leicester NHS Trust | AM/FOI/44829 | Date: 13 August 2021 |
| Walsall Healthcare NHS Trust | FOI/155.21 | Date: 20 July 2021 |
| Buckinghamshire NHS Healthcare Trust | FOI2021-324 | Date: 13 July 2021 |
| Bradford Teaching Hospitals NHS Foundation Trust | FOI/21,168 | Date: 29 June 2021 |
| NHS Greater Glasgow & Clyde | FOI/170/30 | Date: 5 July 2021 |
| NHS Lothian | FOI/5368 | Date: 8 July 2021 |
| Aneurin Bevan University Health Board (Gwent) | FOI/21-256 | Date: 8 July 2021 |
| The Royal Wolverhampton NHS Trust | FOI/8351 | Date: 7 July 2021 |
| Swansea Bay University Health Board | FOIA 21-F-026 | Date: 6 July 2021 |
| Croydon Health Services NHS Trust | FOI/2744 | Date: 2 July 2021 |
| Cambridge University Hospitals NHS Foundation Trust | FOI 2021-06-15 | Date: 14 December 2021 |
| University Hospital Southampton NHS Trust | FOI/7485 | Date: 27 December 2021 |
| Sandwell & West Birmingham Hospitals NHS Trust | F20/0163 | Data never supplied |
| Cardiff & Vale University Health Board | FOI/21.265 | Does not collect ethnicity data |

### 2.2. Data Collection

Each organization was asked to supply:

"Information on the total number of patients with Crohn's disease or ulcerative colitis treated in hospitals in your Trust between 1 January 2015 and 31 December 2019 who were:

1. White British

2. South Asian (Pakistani, Indian, Bangla Deshi, Sri Lankan background)."

Within the UK, patients define their own ethnicity and there is a specific category of "Mixed—White and Asian" in the data set, which the NHS requires all its organisations to complete on each patient. Data were not collected on patients of mixed ethnicity.

Data on the ethnic composition of local communities were obtained from a variety of sources, including health organisations, local councils and nationally collected data. These data sets are based on the same principles as those used in NHS coding.

*2.3. Data Analysis*

The analysis of data was based on the assumption that inflammatory bowel disease (IBD) has the same prevalence in the South Asian community and the White British community in the UK [8–12]. There are weaknesses in this assumption in that recent studies of incidence would suggest that these diseases are commoner in the South Asian population than in the White British community in the UK [12]. In addition, there is clear evidence that the disease can be more severe in the South Asian community [13] and this should lead to more frequent hospitalization. The consequence of these two facts is that the proportion of the South Asian population admitted to hospital could be expected to be greater than that of their White British counterparts.

*2.4. Statistical Analysis*

Comparisons were made between the proportion of hospitalised patients who were South Asian and the proportion who were White British, based on the size of their local communitys using a z statistic (https://www.socscistatistics.com/tests/ztest/default2.aspx (accessed on 11 December 2021)). Where diseases are managed in a comparable fashion, there should be no significant difference between the proportion of South Asian patients with inflammatory bowel disease and the proportion of White British patients with inflammatory bowel disease who were hospitalized locally. If the disease is commoner and more severe in South Asian people, then the proportion hospitalized will be greater than for the White British community, although this difference may not reach statistical significance.

Comparisons were made for each trust or organization on an individual basis. Statistical comparisons were not made between NHS organisations, as the methodology used by Freedom of Information officers to extract information from the data set may be varied.

**3. Results**

Of the 12 English NHS trusts approached, Sandwell and West Birmingham Hospitals NHS Trust failed to provide any data (Tables 1 and 2). The data provided by Buckinghamshire NHS Healthcare Trust suggested considerably lower admission rates for inflammatory bowel disease and was not consistent with data from all other organisations. In Wales, the response from Cardiff and Vale University Health Board was that:

"Cardiff and Vale UHB do not actively capture Ethnicity by default and do not hold these data."

In the case of Aneurin Bevan University Health Board for Gwent and Swansea Bay University Health Board, further analysis was not possible as no specific information on the South Asian community in the relevant areas was available. The only data on ethnic minorities was aggregated together under the heading "Black and Ethnic Minorities" with no breakdown for individual communities.

**Table 2.** Hospital admissions each year over the period 2015–2019 for Crohn's Disease and Ulcerative Colitis.

| Trust | South Asians | White British | Proportion of Patients | South Asian Population | White British Population | Proportion of Population | Z | p |
|---|---|---|---|---|---|---|---|---|
| Coventry and Warwick | 1925 | 12,984 | 0.13 | 64,610 | 283,566 | 0.19 | −18.1 | <0.00001 |
| Leicester | 1424 | 5400 | 0.21 | 174,399 | 849,249 | 0.17 | 8.45 | <0.00001 |
| Wolverhampton | 1763 | 11,523 | 0.13 | 44,960 | 214,642 | 0.21 | −12.66 | <0.00001 |
| Walsall | 495 | 2994 | 0.14 | 43,578 | 220,472 | 0.17 | −3.71 | <0.0002 |
| Bradford | 3062 | 7327 | 0.3 | 144,822 | 364,025 | 0.28 | 2.31 | <0.02 |
| Acute Pennine | 1310 | 12,742 | 0.09 | 184,000 | 616,000 | 0.23 | −38.7 | <0.00001 |
| Croydon | 589 | 2553 | 0.19 | 14,000 | 171,878 | 0.08 | 24.02 | <0.00001 |
| Barking Havering and Redbridge | 691 | 4310 | 0.14 | 168,797 | 187,854 | 0.47 | −47.8 | <0.00001 |
| Buckinghamshire | 9 | 131 | 0.06 | 43,454 | 436,565 | 0.09 | −1.08 | |
| Cambridge | 1025 | 41,595 | 0.02 | 129,945 | 1,202,019 | 0.11 | −50.88 | <0.00001 |
| Southampton | 852 | 21,302 | 0.04 | 20,596 | 219,694 | 0.09 | −22.1 | <0.00001 |
| Lothian | 154 | 1416 | 0.1 | 12,400 | 779,533 | 0.02 | 26.3 | <0.00001 |
| Glasgow and Clyde | 156 | 4577 | 0.03 | 61,200 | 1,004,400 | 0.06 | −7.3 | <0.00001 |
| Gwent | 11–17 | 6667 | 0.003 | | | 0.014 | | |
| Swansea | 14–56 | 1390 | 0.01–0.04 | | | 0.014 | | |

Within these limitations it was possible to undertake a comparison in 10 English NHS trusts and both Scottish organisations. In Leicester, Bradford, Croydon and Lothian the proportion of patients from the South Asian community was significantly greater than the proportion from the local White British community (Table 2), which is consistent with the frequency and severity of the disease in the South Asian community, as a whole in the UK. However, in Coventry, Wolverhampton, Walsall, Acute Pennine Trust in the north-west of England, Barking, Havering and Redbridge, Cambridge, Southampton and Glasgow, South Asian patients were significantly under-represented, indicating significant issues with access to hospital-based healthcare for inflammatory bowel disease.

No data on the detailed ethnic composition of the populations of Gwent and Swansea were available. In addition, due to the small number of South Asian cases in any individual year, it was only possible to identify a range of cases for the quinquennium.

## 4. Discussion

There is on-going evidence of disparate levels of care in inflammatory bowel disease for patients from a South Asian background being underserved by a number of NHS Trusts, Health Boards and comparable organisations. South Asian patients with inflammatory bowel disease are significantly less likely to be admitted to hospital in many areas in England and Scotland where there are large local communities. As the disease is of equal or greater frequency in the South Asian community in the UK [8–12], and of greater severity [13], their rates of hospitalization should be at least equal to, if not greater than is the case for White British patients. This study has shown that Leicester, Croydon, Bradford and the Lothian area are exceptions to this finding. In these areas, a greater proportion of the South Asian community were admitted to hospital [8–13] with inflammatory bowel disease compared to the White British community. Such a finding is consistent with the higher frequency and greater severity of the disease in the South Asian population throughout

the UK. Leicester has taken part in earlier studies, which showed clear evidence that the South Asian community was underserved [1,3]. One of these studies [3] was linked with an educational program directed at clinicians and managers and this may have played a part in the improved rate of hospitalization for South Asian patients with inflammatory bowel disease, demonstrated in the current study.

Differences in levels of delivery of care to South Asian patients first emerged from studies of ulcerative colitis and the provision of biologic therapy to patients with Crohn's disease [14–16]. For example, between 2010 and 2012, South Asian patients with Crohn's disease in Pennine Acute NHS Trust and in Barking, Havering and Redbridge University Hospitals NHS Trust were less likely to receive expensive biologic therapies than White British patients [1]. Subsequent studies confirmed similar problems more extensively throughout England during the period 2010–2015, although the findings were not uniform [1]. As time has progressed, many NHS organisations have limited their collection of data related to ethnicity and expensive therapies. For example, Leicester no longer collects these data and Cardiff and Vale Health Board collect no data on the ethnicity of patients treated at their hospitals. As a consequence, more recent studies have looked at hospital admissions. Between 2014 and 2018, similar disparate levels of care were reported from Northwest Anglia and Essex [2]. In this study, between 2015 and 2019, patients with inflammatory bowel disease were less likely to be admitted to hospital than their White British counterparts. In general, trusts have shown a reluctance to accept any interpretation of their own data which would indicate that they deliver disparate levels of care to various ethnic communities. Such an attitude is consistent with the findings of Salway et al. [4] and means that there is unlikely to be any improvement in the situation in the foreseeable future. However, the present study would suggest that the overall situation in Leicester would appear to have improved and may indicate that effective and on-going educational and awareness programs may have a beneficial impact. Of course, an alternative explanation could be that poorer overall care has led to a greater need for hospital admission. Such an interpretation might have validity in the context of the poorer long-term care in ulcerative colitis that has previously been demonstrated in Leicester [14], together with more limited access to expensive biologic therapies [15]. However, the contemporary situation in Pennine Acute NHS Trust and Barking and Havering and Redbridge University Hospitals NHS would not support this contention [1].

Data collected by Freedom of Information (FOI) techniques have recognised limitations [16,17]. Within a single request, the methodology applied is the same across ethnicities. However, across different FOIs, various methodologies may be used to identify cases, and so comparison between Trusts is inappropriate. This is well illustrated by the response from Buckinghamshire NHS Healthcare Trust, where the number of cases/admissions was very much less than for any other Trust of comparable size. Nevertheless, FOI requests can form the basis for identifying important social issues, which otherwise would go unscrutinised. Such studies can have a direct impact on national policies [18].

A further issue which impacts on ethnicity data is the lack of population data for Trusts. Such data are drawn from related local government structures, but these are not co-terminus with health authorities and so only provide an indication of the likely size of the White British and South Asian populations served. Nevertheless, the magnitude of the differences in provision of care to South Asian and White British patients is such that it cannot be accounted for by such demographic variables.

Disparate levels of care for South Asian patients with inflammatory bowel disease receiving sub-optimal treatment continues to be an issue. Although there is very limited evidence to suggest that the situation may have improved in some trusts, such as University Hospitals of Leicester NHS Trust, over the last decade, such optimism must be treated with caution. For example, a national study, which emanated from Leicester, showed that South Asian patients had poorer access to expensive implantable defibrillators than their White British counterparts [19]. In addition, there appears to be a significant decline in the nature and type of data on ethnicity collected in England by NHS trusts. The situation in Wales is

particularly disturbing, where data on individual ethnic minority groups are aggregated into a single "Black and ethnic minorities" coding, limiting any analysis.

The findings from this study reinforce the need to develop effective and independent systems for collecting data on the provision of care by Trusts, Health Boards and similar organisations, when there is on-going failure to achieve the objectives of the NHS:

> "to promote equality through the services it provides and to pay particular attention to groups or sections of society where improvements in health and life expectancy are not keeping pace with the rest of the population." [20]

It is critical to introduce effective policies which will alter the in-built inertia to change within such organisations.

**Author Contributions:** Both authors contributed equally to design, collection of data, analysis and writing of the paper. All authors have read and agreed to the published version of the manuscript.

**Funding:** This research received no external funding.

**Institutional Review Board Statement:** The study was conducted on freely available data obtained by Freedom of Information requests.

**Informed Consent Statement:** Not applicable.

**Data Availability Statement:** All original data can be obtained from the relevant Trusts quoting the FOI number.

**Conflicts of Interest:** The authors declare no conflict of interest.

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
