# Peer review of "Evidence of On-Going Disparate Levels of Care for South Asian Patients with Inflammatory Bowel Disease in the United Kingdom during the Quinquennium 2015–2019"

_gastrointestdisord, doi:10.3390/gidisord4010002_

Round 1

Reviewer 1 Report

Thank you for the opportunity to peer review this manuscript. My overall recommendation is to reject. Nevertheless, I provide some suggestions below so you can enhance your manuscript for the next journal you submit to.

Regarding the abstract. It lacks one or two background sentences (i.e. disparate care between ethnicities).

When you say "In Leicester, Bradford, Croydon and Lothian the proportion of patients from the South Asian community was significantly greater than the proportion in the local community, which is consistent with the epidemiology and severity of the disease in the South Asian community." are you saying that you expect more South East Asians to be hospitalised than caucasians because they have more severe disease? If so, I do not think you have elucidated this enough in the manuscript.

The list you have under methods would probably go better in a Table.

You need more subheadings in methods. Consult the "Strobe checklist", easily findable on google, to see which things you should describe in your manuscript. Examples of subheadings you may want to seriously consider having are: Participants, Data collection, Statistical Methods among others.

At line 82, I think you need to describe this better, perhaps summarise in a Table or supplementary material. 

What statistical software was used? Was a statistician consulted?

Table 2 needs straightening. 

Table 2 can probably go before Table 1 as that makes more sense as you are describing where you got your data from whereas Table 1 is describing what your obtained data was. Table 2 should probably be put in methods and/or as supplementary material.

Is Gwent the same as Aneurin Bevan University Health Board? If so, you may want to make the main text consistent with the table.

If someone has 1 South asian parent and 1 caucasian British parent, how are they categorised in the 2 datasets? Is the way of coding ethnicity the same in the hospitals as the general population?

In your discussion, your first paragraph is too long.

You seem to have limitations in your first large paragraph, this should be a bit later on.

Your first paragraph should more or less say what your results were and how they aligned with your aims/hypotheses.

At line 164-166, you say "the overall situation in Leicester would appear to have improved, such that a significantly greater proportion of South Asian patients are admitted to hospital with inflammatory bowel disease, consisent with its greater prevalence in this community". Are you trying to say SA patients have high rates of IBD in Leicester? If so, what is your citation?

Your hypothesis/what you expect to find needs to be more explicitly stated in the final paragraph of the introduction.

Author Response

1. A few sentences have been added to the abstract about disparate care between ethnicities.

2. References to the greater frequency of  inflammatory bowel disease in the South Asian community and the greater severity in that community have been added throughout the text. As a result one would expect a greater proportion of South Asians to be hospitalised than White British patients.

3. The list has been converted into a Table. The Table is now Table 1 and resited in the text.

4. Subheadings have been added to the Methods section, including:: Participants, Data collection and Data analysis.

5. The source of the statistical software is now included.  A statistician was not consulted as one of the authors is an epidemiologist.

6. Table 2, which has become Table 1 has been straightened. 

7. It is now made clear that Gwent is the same as Aneurin Bevan University Health Board.

8. NHS coding has a specific categories for patients of mixed ethnicities. The system is also used for general population data and statements concerning this are now included in the Methods.

9. The first paragraph of the Discussion has been shortened and deals with the findings of this study and its relationship to the hypothesis under investigation..

10. Limitations are dealt with in a later paragraph now.

11.  The statement "the overall situation in Leicester would appear to have improved, such that a significantly greater proportion of South Asian patients are admitted to hospital with inflammatory bowel disease, consistent with its greater prevalence in this community" has been changed to say that the figures are now consistent with the greater frequency and severity of the disease, which is true for the whole of the UK and not specifically Leicester. It has, of course, been shown in Leicester as well. References to the frequency and severity of the disease are included earlier in the paper (References 8 - 13)

12. The hypothesis is now stated explicitly in the introduction.

Reviewer 2 Report

Thanks for the opportunity to review.

The manuscript will need to be rewritten and changed to Disparity in Care rather than Disparate Care.

The methods need to be made clearer to support findings,

Author Response

  1. The manuscript has been extensively rewritten.
  2. In particular, the Methods section has been rewritten with sub-headings and hopefully greater clarity.
  3. The title has been changed so as to use Disparity in Care rather than Disparate Care.

Round 2

Reviewer 1 Report

Thank you for the opportunity to peer review this revision. In the initial review, I recommended this be rejected but the editor informed me that the other reviewer and academic editor wanted to give you a chance to revise and so I have been asked to re-review this.

Thank you for your responses. I respond systematically below.

  1. Thank you for your change.
  2. I see what you have added to the discussion regarding IBD frequency in Asians. You need to consider having a sentence in the introduction to set the scene for the hypothesis in this regard. On that note, the hypothesis sentence should be the last one in the introduction, not the first. Your introduction needs some reordering with a logical buildup to the hypothesis.
  3. Thank you for the change.
  4. Thank you for the change.
  5. Thank you for the change.
  6. Thank you for the change.
  7. Thank you for the change.
  8. Thank you for the change.
  9. Thank you for the change.
  10. Thank you for the change.
  11. Thank you for the change.
  12. As I said earlier, your hypothesis should be closer to the end of the introduction as opposed to the first sentence.

Author Response

2. The introduction has been restructures and a sentence included about IBD frequecy in the South Asian community.

12. The hypothesis is now at the end of the introduction.

Reviewer 2 Report

The manuscript title was changed as suggested but throughout the paper, disparate care and disparate levels of care are used. The terminology needs to be consistent. 

Author Response

  1. The introduction has been revised.
  2. "Levels of disparate care" are now used throughout the manuscript

Round 3

Reviewer 2 Report

I can not recommend the paper be published using disparate care. As stated in my previous review, it should be disparity in care throughout the paper. 

Author Response

1 Subheadings have now been added to the RESULTS as used in the METHODS section.
2. The table has been reformatted so proportion is on one line
3. I believe the references are in the format required by the journal
4."Disparity in care" or "disparities in care"  is used throughout the paper,.